# Hierarchical Channel-spatial Encoding for Communication-efficient Collaborative Learning

**Qihua Zhou**[1], **Song Guo**[1*], **Yi Liu**[1], **Jie Zhang**[1], **Jiewei Zhang**[1],
**Tao Guo**[1], **Zhenda Xu**[1], **Xun Liu**[1], **Zhihao Qu**[2]

[1]Department of Computing, The Hong Kong Polytechnic University
[2]School of Computer and Information, Hohai University
{csqzhou,csyiliu,csjwzhang}@comp.polyu.edu.hk
{jieaa.zhang,cocotao.guo,jackal.xu,compxun.liu}@connect.polyu.hk
song.guo@polyu.edu.hk, quzhihao@hhu.edu.cn

## Abstract

It witnesses that the collaborative learning (CL) systems often face the performance bottleneck of limited bandwidth, where multiple low-end devices continuously generate data and transmit intermediate features to the cloud for incremental training. To this end, improving the communication efficiency by reducing traffic size is one of the most crucial issues for realistic deployment. Existing systems mostly compress features at pixel level and ignore the characteristics of feature structure, which could be further exploited for more efficient compression. In this paper, we take new insights into implementing scalable CL systems through a hierarchical compression on features, termed *Stripe-wise Group Quantization* (SGQ). Different from previous unstructured quantization methods, SGQ captures both channel and spatial similarity in pixels, and simultaneously encodes features in these two levels to gain a much higher compression ratio. In particular, we refactor feature structure based on inter-channel similarity and bound the gradient deviation caused by quantization, in forward and backward passes, respectively. Such a double-stage pipeline makes SGQ hold a sublinear convergence order as the vanilla SGD-based optimization. Extensive experiments show that SGQ achieves a higher traffic reduction ratio by up to $15.97\times$ and provides $9.22\times$ image processing speedup over the uniform quantized training, while preserving adequate model accuracy as FP32 does, even using 4-bit quantization. This verifies that SGQ can be applied to a wide spectrum of edge intelligence applications.

## 1   Introduction

Recent years have seen great prospects of deploying vision tasks on tiny edge devices by using their always-on microprocessors, embedded sensors and neural chips. Considering the realistic environment that new data is continuously generated on user devices that cannot be aggregated at once due to the privacy and energy concerns, it comes to the rise of collaborative learning (CL) paradigm [1, 22, 15]. Considering the resource constraints in CL systems, it is expected to partition the models between multiple edge devices and the cloud, and coordinate the two sides during the training procedure [32, 7]. In the forward pass of CL, the shallow *cut layers* deployed on edge devices are used for low-level feature extraction and the intermediate features will be transmitted to cloud for the subsequent processing on deep *remaining layers*. Also, in the backward pass, the cloud returns the derivative flows of the transmitted features to the devices for model updating. By adopting such a learning paradigm, devices can update parameters and evolve models continuously.

---

*Corresponding author

36th Conference on Neural Information Processing Systems (NeurIPS 2022).

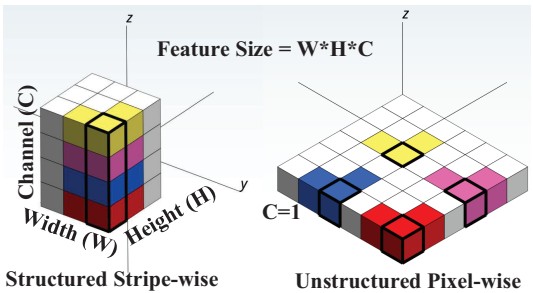

Figure 1: Visualization of the structured stripe-wise quantization (the left, ours, maintaining 4-channel structure) v.s. the unstructured pixel-wise quantization (the right, *e.g.,* UQ [41, 14, 40], flattening original structure as 1 channel), which are with the $3 \times 3$ and $6 \times 6$ pixel plane, respectively. The batch size is set as $1$ for illustration convenience.

**Prior work and limitations.** In general, deploying an efficient CL system needs to address communication deficiency caused by limited bandwidth [15, 27], where reducing feature size is the key to improve communication performance. Existing communication-optimized methods, including progressive model slicing (*e.g.,* CLIO [12]), matrix factorization (*e.g.,* Poseidon [37]) and top-k filter (*e.g.,* Async-opt [3]), either hold an insufficient traffic reduction ratio or may degrade the model accuracy if compressing features. Besides, existing quantization methods cannot directly be adopted to feature compression as they are specifically designed for weights [19, 41, 38] or activations [4], which hold different distribution characteristic from the features [8]. Simply applying these methods cannot fundamentally reduce feature size as verified by our experiments (see §4.3), which is not applicable to realistic CL systems. This motivates us to design a new quantization scheme that provides a desired compression ratio on feature maps while not sacrificing the model quality.

**Observations and challenges.** Actually, the feature regions often hold specific pixel similarity across channels when the kernels aim at extracting interrelated features, especially in shallow layers [8, 23, 39, 26, 2]. We could leverage this property to quantize each pixel along all the channels in a structured manner (*i.e.,* the stripe wise in Figure 1), rather than the "flat" perspective (*i.e.,* the pixel wise in Figure 1) of *Uniform Quantization* (UQ) [41, 14, 40]. However, by analyzing the value distribution of each channel inside the feature, we observe that some channels hold quite different features when the corresponding filters are orthogonal to each other. This indicates that simply adopting the vanilla *Product Quantization* (PQ) [30, 5] along the whole channel dimension will introduce a significant representation error on features and finally degrade the model accuracy. We need to reorganize the features into groups based on their channel-level similarity, instead of treating the features as a whole or roughly partitioning them into successive subsets. Therefore, capturing such channel-dimension structured information is the key to fundamentally compress feature size, which is often omitted by conventional quantization methods designed for parameters, activations or gradients. As to each group, we need to find a collection of representative pixels, each of which can replace other pixels similar to it.

**Our solution.** We achieve the above target by proposing the *Stripe-wise Group Quantization* (SGQ, §2.1) method, which captures both channel and spatial-level similarity in pixels, and hierarchically encodes the features in these two levels to achieve a much higher compression ratio. Specifically, we introduce the *Channel-attention Grouping* (CAG, §2.2) block to measure the per-channel significance and reorganize the entire features along the channel dimension into several groups, each of which holds similar inter-channel texture and pixel intensity reflected by channel significance. In each group, we employ K-means to divide the feature maps into several clusters based on pixel similarity along the stripe (*i.e.,* vector) and represents all the pixels belonging to each cluster by its centroid. As all the pixels are clustered along the channel dimension simultaneously, more volume of the features can be quantized over the conventional unstructured pixel-wise quantization, yielding a much smaller pixel plane. Thus, given the same quantization bits, SGQ could achieve a higher compression ratio, especially for features with a large channel number, which is common in modern CNNs. For example, as to the 4-channel $3 \times 3$ feature map shown in Figure 1, SGQ provides a $4\times$ higher compression ratio over the unstructured pixel-wise UQ using same quantization bits. We implement our idea into a novel CL framework, which could automatically insert the SGQ block into different models and adjust clustering hyper-parameters according to model characteristics in

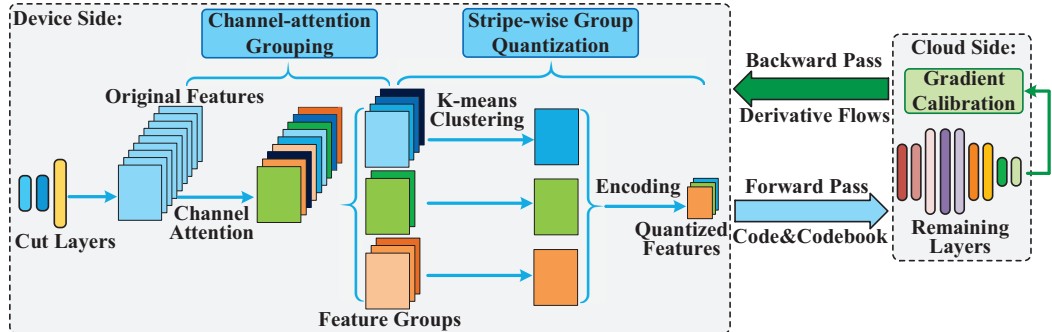

Figure 2: The overview of our framework across edge devices and the cloud.

the forward pass. To preserve model accuracy, we also design the *Gradient Calibration* (GC, §2.3) module to adjust the corresponding gradients of quantized features and CAG block in the backward pass, which holds a sublinear convergence order as the vanilla SGD. This double-pass pipeline makes SGQ a communication-efficient method to enable model evolution on multiple edge devices without sacrificing much model accuracy as the FP32 training.

**Advantages and contributions.** Evaluation based on NVIDIA Jetson Nano [24] and HUAWEI Atlas 200DK [13] shows that SGQ could effectively alleviate the communication overhead for feature transmission and provide comparable model quality as the original FP32 training, supporting various CNN models. Thus, SGQ can serve as a lightweight module for the resource-constrained CL environments. Overall, the key contributions of our work are as follows:

- **Scalable collaborative learning framework:** We propose a scalable CL framework that enables model evolution on multiple edge devices and match the requirements of continuous analytics. It will be open-source[2] and will support commodity edge devices (*e.g.,* NVIDIA Jetson Nano and HUAWEI Atlas 200DK), thus can be easily deployed in realistic scenarios.

- **General feature compression method:** We address the communication bottleneck by capturing the structured pixel similarity in both channel and spatial levels, and propose the SGQ method (§2.1) to hierarchically encode the features in these two levels for a much higher compression ratio over existing methods. SGQ can serve as a general quantization block supporting different CNN models, without sacrificing much model accuracy as the FP32 training.

- **Efficient convergence order:** We formulate the impact of quantized features on the training process and present the theoretical analysis to bound the gradient deviation in backward pass, making SGQ-based training holds a sublinear convergence order (§3) as the vanilla SGD algorithm.

To the best of our knowledge, the proposed SGQ is the first general traffic reduction method exploiting the quantization feasibility in both spatial and channel levels for building communication-efficient CL systems. It achieves a much higher feature compression ratio over the pertinent existing methods while not sacrificing model accuracy even in 4 bits, thus providing great advantages for real-world implementation. We believe SGQ could constantly contribute to the further development of edge intelligence applications.

## 2  Methodology

The key of our collaborative learning framework is to quantize feature maps for traffic reduction and fetch the corresponding gradients for model updating in forward and backward passes, respectively. As shown in Figure 2, this target is resolved by the *Stripe-wise Group Quantization* (SGQ) based on *Channel-attention Grouping* (CAG) block and the *Gradient Calibration* (GC) method.

---

[2]Source codes will be shared at Github after the double-blind review.

## 2.1 Stripe-wise Group Quantization

SGQ is used for compressing the feature map size at the end of device's cut layers by converting the FP32 values to low-bit format (*e.g.,* INT8). It contains two key steps: (1) feature discretization and (2) pixel encoding.

**Step #1: Feature discretization.** In this step, all the pixels of the feature maps belonging to a same group are categorized into several clusters, which transfers the "continuous" FP32 values to the discrete ones. The number of clusters is called the quantization level and directly impacts the data representation precision. Pixels will be covered by $2^n$ clusters if we use the $n$-bit quantization, where $n$ is usually set as 8 or 4.

This procedure can be handled by the K-means clustering under given bits. Considering the computational overhead of K-means, we need to downsamples the feature maps first and generate the K-means clustering model based on these samples, instead of the whole feature maps. Thus, the overhead can be restricted within $6\%$ of the forward pass time by using the suggested 4-bit quantization (details are in §4.5). Given the feature maps $\mathbf{X} = \{\boldsymbol{x}_1, \boldsymbol{x}_2, \cdots, \boldsymbol{x}_m\}$ generated at the end of device's cut layers, we categorize $\mathbf{X}$ into $k$ ($k = 2^n$) clusters $\{c_1, c_2, \cdots, c_k\}$, where each $\boldsymbol{x}_i$ is a pixel of $\mathbf{X}$ and the corresponding clustering centroid matrix is $\mathbf{U} = \{\boldsymbol{u}_1, \boldsymbol{u}_2, \cdots, \boldsymbol{u}_k\}$. Therefore, the cost function of K-means clustering can be formulated as:

$$J = \sum_{i=1}^{m} \sum_{j=1}^{k} r_{i,j} \|\boldsymbol{x}_i - \boldsymbol{u}_j\|_2^2, r_{i,j} \in \mathbf{R}_{m \times k}, \tag{1}$$

$$r_{i,j} = \begin{cases} 1, & \boldsymbol{x}_i \in c_j, \\ 0, & else. \end{cases} \tag{2}$$

where $\mathbf{R}$ is the pixel mapping matrix generated by K-means clustering, reflecting whether $\boldsymbol{x}_i$ belongs to $c_j$. For each pixel $\boldsymbol{x}_i$, we can calculate its clustering centroid $y_i$ as:

$$y_i = \underset{j \in \{1, 2, \cdots, k\}}{\arg\min} \|\boldsymbol{x}_i - \boldsymbol{u}_j\|_2. \tag{3}$$

By restricting the partial derivatives of Eq. (1) as 0, we can figure out the latest centroids as:

$$\boldsymbol{u}_k = \frac{\sum_{i=1}^{m} r_{i,k} \boldsymbol{x}_i}{\sum_{i=1}^{m} r_{i,k}}. \tag{4}$$

The above procedure will repeat until all the centroids are stable enough to form the clustering model and we finally formulate the feature discretization as $\mathbf{Y}_{m \times 1} = D(\mathbf{X}_{m \times C})$, where $D$ represents the discretization function that maps each pixel $\boldsymbol{x}_i$ to the one-hot cluster label $y_i$ ($y_i \in \mathbf{Y}$).

**Step #2: Pixel encoding.** This step represents each pixel by the centroid of the cluster it belonging to, such that all the pixels are encoded as the unique index of the corresponding clustering centroid, which can be covered by $n$ bits. The entire feature maps is compressed to $\frac{n}{32}$ of the original FP32 size. For brief, such encoding procedure that reflects the mapping function between the original pixel and the centroid is called the codebook.

This procedure can be handled by a series of matrix transformation. Given the pixel mapping matrix $\mathbf{R}_{m \times k}$ and the clustering centroid matrix $\mathbf{U}$, each row $r_j$ of the pixel mapping matrix $\mathbf{R}$ represents the cluster label of $y_j$ in the one-hot form. Therefore, we can get the transformation relation between $\mathbf{Y}_{m \times 1}$ and $\mathbf{R}_{m \times k}$ as $\mathbf{R} = \text{onehot}(\mathbf{Y})$. Then, the pixel encoding process can be formulated as:

$$Q(\boldsymbol{x}_i) = \sum_{j=1}^{k} r_{i,j} \boldsymbol{u}_j, \boldsymbol{x}_i \in \mathbf{X}, \tag{5}$$

$$Q(\mathbf{X}) = \mathbf{R} \cdot \mathbf{U} = \text{onehot}(\mathbf{Y}) \cdot \mathbf{U}. \tag{6}$$

By approximating the one-hot $\mathbf{Y}$ via the *softmax* function, we can get each encoded pixel $\hat{\boldsymbol{x}}_i$ as:

$$\hat{\boldsymbol{x}}_i = Q(\boldsymbol{x}_i) = \frac{\sum_{j=1}^{k} \boldsymbol{u}_j e^{-(\boldsymbol{x}_i - \boldsymbol{u}_j)(\boldsymbol{x}_i - \boldsymbol{u}_j)^\top}}{\sum_{j=1}^{k} e^{-(\boldsymbol{x}_i - \boldsymbol{u}_j)(\boldsymbol{x}_i - \boldsymbol{u}_j)^\top}}, \hat{\boldsymbol{x}}_i \in Q(\mathbf{X}). \tag{7}$$

where the distance between $x_i$ and $u_i$ is minimized. In summary, the above two steps constitute the basic function of SGQ and we can get the final quantized feature map $Q(\mathbf{X})$.

**Traffic analysis.** Different from the existing quantization methods that flats the entire feature maps and conduct quantization in the pixel wise, SGQ quantizes all the pixels along with the channel dimension (*i.e.,* the stripe wise), thus providing a much higher compression ratio of feature maps. As shown in Figure 1, we make a comparison of the traffic size by using conventional *Uniform Quantization* (UQ) [41, 20, 14] and SGQ, under the same quantization bits.

As the original model is split between edge devices and the cloud, the cloud needs both quantized feature maps and codebook to recover the intermediate results generated by the cut layers, so as to promote the computation of remaining layers. The major network traffic is dominated by the quantized feature maps and codebook. Note that the codebook is still represented in the FP32 data format while the quantized feature maps only requires $n$ bits. Given $G$ groups, each of which holds $C_i$ channels, the traffic size $S_{SGQ}$ by using SGQ can be described as:

$$S_{SGQ} = \sum_{i=1}^{G} (\underbrace{n \cdot WH}_{feature} + \underbrace{32 \cdot 2^n \cdot C_i}_{codebook}), \tag{8}$$

where $W$, $H$ represent the with and height of the pixel plane, respectively. Correspondingly, the UQ can be regarded as a special version that entire feature map is flatted as 1 channel, thus the traffic size $S_{UQ}$ based on UQ is described as:

$$S_{UQ} = \underbrace{n \cdot WH \cdot \sum_{i=1}^{G} C_i}_{feature} + \underbrace{32 \cdot 2^n}_{codebook}. \tag{9}$$

To make SGQ generate less traffic size, we require $S_{SGQ} < S_{UQ}$ and this inequation can be simplified as:

$$\frac{n \cdot WH}{2^{n+5}} > \underbrace{\frac{\sum_{i=1}^{G} C_i - 1}{\sum_{i=1}^{G} C_i - G}}_{\approx 1}. \tag{10}$$

As $G$ is usually far smaller than $\sum_{i=1}^{G} C_i$ in practice, Eq. (10) is easy to satisfy in common CNN models. For example, if we cut ResNet18 at CONV1 with 4-bit and 10-group SGQ, we will have the quantitative relation that $\frac{4 \times 112 \times 112}{2^9} \gg \frac{64-1}{64-10}$. Therefore, SGQ can provide a much higher compression ratio over conventional UQ, thus effectively reducing traffic size of feature transmission.

## 2.2 Channel-attention Grouping Block

Recall that SGQ makes feature discretization based on K-means clustering, which closely replies on the distance measurement between pixels. Actually, not all channels are equally important to the representation of feature maps. Assigning each channel with equal distance weighting cannot well capture the characteristics of the entire feature map. A natural idea is to precisely reflect per-channel significance and "pay attention" to the most significant channels for more efficient clustering. More seriously, channels may hold similar texture or orthogonal to each other. Simply conducting the vanilla *Product Quantization* (PQ) [30, 5] to pixels along the channel dimension will introduce a significant representation error because it is hard or even impossible for PQ to find the proper cluster centroids that can replace other pixels, thus finally degrading the model accuracy. We need to reorganize the features into groups based on their channel-level similarity, instead of treating the features as a whole or roughly partitioning them into successive subsets.

Consequently, we design the *Channel-attention Grouping* (CAG) block to capture the channel-dimension structured information for more precise clustering. As shown in Figure 2, the key of CAG block is to form a one-dimension vector with $C$ elements, each of which represents the distance weighting of corresponding channel. This function can be abstracted as a series of affine transformation $A(\mathbf{X})$ that converting the original feature maps into the $C$-element channel significance vector $\mathbf{V}_{1 \times C}$, *i.e.,* $\mathbf{V} = A(\mathbf{X})$.

In practice, we could employ a two-layer fully-connected network to approximate this procedure, which is inspired by the self-attention mechanism [6, 34, 11, 35]. At beginning, we use common downsampling methods (*e.g.,* AVG pooling) to shrink each channel of the feature map into one pixel, thus the feature map is compressed as a $C$-element vector $\mathbf{V}_{1 \times C}$. Then, the two-layer fully-connected block conducts an affine transformation to extract channel-level significance and the transformed vector $\mathbf{V}$ represents the distance weighting of each channel. We can use this vector to adjust the distance measurement for K-means, which is described as:

$$\sum_{i=1}^{m} \sum_{j=1}^{k} v_i \cdot r_{i,j} \|\boldsymbol{x}_i - \boldsymbol{u}_j\|_2^2, v_i \in \mathbf{V}. \tag{11}$$

Consequently, the CAG block helps SGQ extract the most informative channels and preserves the spatial characteristics of feature maps after pixel encoding.

## 2.3 Gradient Calibration

As the feature maps generated by the cut layers have been quantized in the forward pass, we need to adjust the corresponding derivative flows in the backward pass, so as to figure out the correct gradients of both feature maps and CAG block for preserving model convergence. We achieve this target by designing the *Gradient Calibration* (GC) module. The theoretical analysis of model convergence by employing GC module will be discussed in §3. Here, we will highlight the gist of how GC works to feature maps and CAG block, respectively.

**Gradients of quantized feature maps.** Based on the formulation of SGQ function in Eq. (7), we can calculate the theoretical gradients of quantized feature maps as:

$$\frac{\partial Q(\boldsymbol{x}_i)}{\partial \boldsymbol{x}_i} = \frac{2 \sum_{j=1}^{k} \sum_{l=1}^{k} \boldsymbol{u}_j^\top (\boldsymbol{u}_j - \boldsymbol{u}_l) p_j p_l}{\sum_{j=1}^{k} \sum_{l=1}^{k} p_j p_l}, \tag{12}$$

$$\approx 2\hat{\boldsymbol{x}}_i \cdot \mathbf{U}^+ \cdot \mathbf{U}_o - 2\hat{\boldsymbol{x}}_i^\top \cdot \hat{\boldsymbol{x}}_i \tag{13}$$

where $p_j = e^{-(\boldsymbol{x}_i - \boldsymbol{u}_j)(\boldsymbol{x} - \boldsymbol{u}_j)^\top}$ and $\mathbf{U}^+$ is the pseudoinverse of $\mathbf{U}$. Note that $\mathbf{U}_o$ is the dot product matrix, where the i-th element is $\boldsymbol{u}_i^\top \boldsymbol{u}_i$. As the dot product of clustering centroid matrix $\boldsymbol{u}_i$ may introduce extra computational overhead in Eq. (13), we can store these clustering matrices and conduct the calculation on the cloud side. Then, the cloud only needs to return the intermediate derivative flow of $\frac{\partial Q(\boldsymbol{x}_i)}{\partial \boldsymbol{x}_i}$ to edge devices during the backward pass, which will not yield much traffic. Full derivation of Eq. (13) is in the supplementary material of § B.

**Gradients of CAG block.** The CAG block serves as an independent branch to measure the channel-level significance and the transformed vector $\mathbf{V}$ is merged to the SGQ method for distance calculation. Therefore, the gradient calculation of CAG block involves the operations of branch merging and duplication. Based on the derivative flow generated by the backward pass of quantized feature maps, the gradients of branch merging is described as:

$$\frac{\partial A(\mathbf{X})}{\partial \mathbf{X}} = \frac{\partial A(\mathbf{X})}{\partial Q(\mathbf{X})} \cdot \frac{\partial Q(\mathbf{X})}{\partial \mathbf{X}} = \mathbf{V} * \frac{\partial Q(\mathbf{X})}{\partial \mathbf{X}}, \tag{14}$$

where $*$ denotes the element-wise multiplication that broadcasts each $v_i \in \mathbf{V}$ to the corresponding position of $\frac{\partial Q(\mathbf{X})}{\partial \mathbf{X}}$. Also, the gradients of branch duplication can be regarded as the sum of derivative flows. Thus, the final gradients of CAG block is described as:

$$\frac{\partial A(\mathbf{X})}{\partial \mathbf{X}} = \mathbf{V} * \frac{\partial Q(\mathbf{X})}{\partial \mathbf{X}} + \frac{\partial Q(\mathbf{X})}{\partial \mathbf{X}}. \tag{15}$$

Note that CAG block only holds a slight computational overhead (less than 2%) in both forward and backward passes. The detailed analysis will be discussed in §4.5.

## 3 Convergence Analysis

**Theorem 1.** *Assuming each pixel follows $\|\boldsymbol{x}\|_2 \leq B$, the upper bound of the approximate gradients can be described as $\|\frac{\partial Q(\boldsymbol{x})}{\partial \boldsymbol{x}}\|_2 \leq 2B^2$. Meanwhile, given $m$ pixels and $C$ channels, the upper bound*

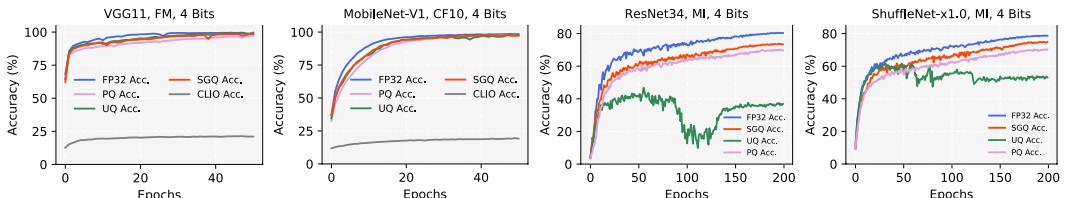

Figure 3: Comparison of convergence curves using different benchmarks and baselines.

Table 1: Summary of average model accuracy (%) using 4-bit compression, compared with FP32.

| Method | VGG11, FM | MobileNet-V1, CF10 | ResNet34, MI | ShuffleNet-x1.0, MI |
|---|---|---|---|---|
| FP32 (Upper Bound) | 97.55 | 94.74 | 80.31 | 78.73 |
| UQ | 95.12 | 92.41 | 36.89 | 53.15 |
| PQ | 95.94 | 92.67 | 69.61 | 70.16 |
| CLIO | 21.02 | 19.16 | 13.06 | 11.10 |
| **SGQ** | **96.57** | **93.45** | **74.37** | **74.86** |

*of real gradients is described as $\frac{C}{m} \le ||\boldsymbol{g}_x||_2 \le C$. Therefore, the approximate gradients generated by GC follows $||\frac{\partial Q(\boldsymbol{x})}{\partial \boldsymbol{x}}||_2 \le \frac{2B^2 m}{C}||\boldsymbol{g}_x||_2$.*

Theorem 1 indicates that the $\ell_2-$norm of the gradient derived under quantized feature map is bounded by the product of a constant and the $\ell_2-$norm of the original gradient. Under the widely applied assumption of bounded gradient and the conclusion of Doublesqueeze [31, 29], the proposed SGQ approach holds the same convergence order as the SGD method without quantization on feature maps. The detailed proof can be found in the supplementary material of § C.

## 4 Experiments

### 4.1 Experiment Setting

**Devices.** To match the edge environment, we evaluate SGQ on two types of devices: (1) NVIDIA Jetson Nano series [24], and (2) HUAWEI Atlas 200DK [13], both of which are connected to the NVIDIA RTX 2080Ti server through 10GbE network.

**Benchmarks.** Our benchmarks are image classification tasks based on the training of AlexNet [18], VGG-11 [28], ResNet-18/34 [9], ShuffleNet-V2-1.0x/0.5x [21], and MobileNet-V1 [10], with the CIFAR-10/100 (CF10/100) [17], Fashion MNIST (FM) [36] and mini-ImageNet (MI) [33] datasets. As to MI, the batch size is 32 with the SGD optimizer. As to CF and FM, the batch size is 100 with the Adam [16] optimizer. All of these benchmarks are implemented via PyTorch-1.7.1 [25].

**Baselines.** We inspect the proposed SGQ method with four pertinent baselines: (1) the vanilla full-precision training (FP32), (2) the uniform quantization (UQ) [41], (3) the product quantization (PQ) [30] and (4) the progressive-slicing CLIO [12], which uses the top $\frac{n}{32}$ slices corresponding to the same compression ratio of $n$-bit UQ. Note that our SGQ under same representation bits provides a much higher compression ratio over $\frac{n}{32}$. The details of traffic saving are in §4.3.

### 4.2 Convergence Efficiency

We inspect the training convergence curves of SGQ and other baselines. As shown in Figure 3, we can observe that SGQ (red) achieves the highest accuracy over other baselines using different benchmarks, verifying SGQ is a general feature compression method and can be applied to the training of most CNNs. Even training on the small-scale FM and CF10 datasets, the CLIO-based training (gray) cannot converge under the same compression ratio, although its Top-k like variants are widely used for compressing parameters and gradients in distributed model training. This phenomenon indicates that simply dropping a certain proportion of pixels or channels in feature maps will cause significant information loss and finally destroys the training convergence. Meanwhile, UQ (green) cannot preserve an acceptable model accuracy when training on the large-scale MI datasets, with severe fluctuation along the epochs. This is because UQ needs to flat the entire feature maps before

quantization, thus losing the spatial characteristics of the pixels among different channels. Inversely, both SGQ and PQ (pink) can maintain good model accuracy as their quantization schemes well maintain the feature structure after pixel encoding. However, our SGQ achieves a faster convergence rate and holds higher model accuracy over the PQ method, because the channel-agnostic PQ cannot capture the difference of orthogonal channels and roughly quantizing features via successive division will introduce quantization errors. For a more straightforward comparison, we summarize the average model accuracy of different method using the same compression ration of 4 bits in Table 1. It is clear that SGQ outperforms other baselines in different training configurations and does not much sacrifice the model accuracy as FP32 does. Remarkably, Figure 3 and Table 1 only display the results of 4-bit training, our SGQ method can also achieve good training performance even in extremely low quantization bits (*e.g.,* 3 bits), which well matches the resource constraints of edge devices.

## 4.3    Traffic Saving

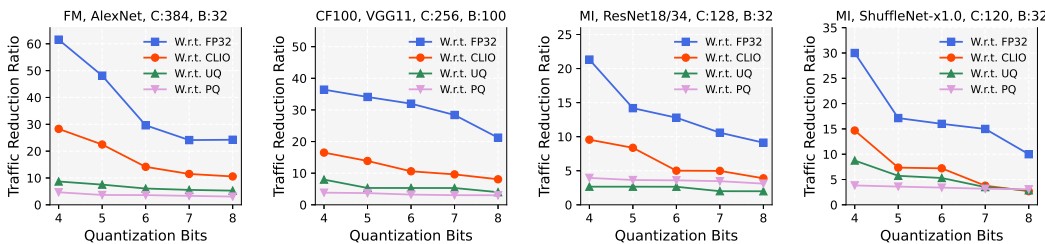

Figure 4: Average traffic reduction ratio by using SGQ.

As shown in Figure 4, by presenting two pertinent cases, we highlight SGQ's traffic reduction ratio, over the FP32 training, CLIO (using top 25% slices) and UQ, where

$$\text{Traffic Reduction Ratio} = \frac{\text{Baselines's Traffic}}{\text{SGQ's Traffic}}.$$

We did not compare the results under 2 and 3 bits because UQ and CLIO cannot achieve stable model accuracy in such configuration. With the hierarchical channel-spatial encoding, SGQ can compress feature maps more efficiently, thus providing prominent traffic reduction ratios over the baselines. Specifically, we can observe that SGQ's reduction ratio increases when (1) using lower quantization bits (*e.g.,* 4 bits), (2) cutting at layers with more channels (*e.g.,* C:256), and (3) using larger batch size (*e.g.,* B:100). In these three cases, the feature maps follow the long strip shape, where conducting channel-level clustering based on lower bits will yield smaller quantized features and corresponding codebook, thus significantly reducing the traffic size for transmitting feature maps. However, UQ cannot quantize the pixels along the channel dimension simultaneously, limiting its effectiveness of realistic deployment. Such a degree of reduction ratio makes SGQ an adequate feature compression method for the low-bandwidth edge environment. Overall, the superiority of SGQ over existing methods can be quickly understood by checking the results in Figure 5. SGQ explicitly outperforms existing methods by achieving better trade-off between model accuracy and traffic size.

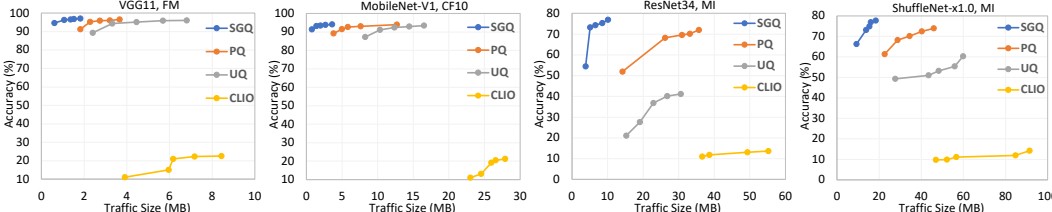

Figure 5: SGQ significantly outperforms existing methods in both model accuracy and traffic size.

## 4.4    Ablation Study

**Impact of quantization bits.** We also compare the final model accuracy of SGQ by using different quantization bits and training benchmarks in Figure 6. As the quantization bits directly impact

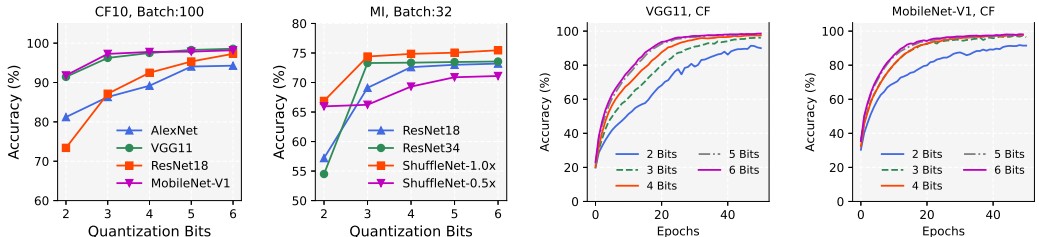

Figure 6: Average model accuracy by using different bits, where the 4-bit configuration is recommended for SGQ's deployment in practice. The detailed convergence curves of each model under different bits are provided in the supplementary material of § D.

the cluster numbers in feature discretization, lower bits will lead to more accuracy degradation. In most cases, SGQ achieves acceptable accuracy under different bits, even for the 2-bit training. (*e.g.,* MobileNet and ShuffleNet). This property makes SGQ can effectively save traffic size for transmitting feature maps. Note that the training quality is also related to the splitting position of cut layers. Generally, given the same quantization bits, cutting at the shallower layers holding relatively larger feature section (*i.e.,* the product of width and height) but smaller channel number can achieve higher model accuracy. However, in the extreme case that features are with wide channels (*e.g.,* 512) while with small pixel plane (*e.g.,* $4 \times 4$), SGQ's performance may degrade as there is not enough clustering space to conduct feature discretization. Although we can reshape the feature maps to reduce channels and expand the pixel plane, the compression ratio of feature size will also decrease. As the 4-bit SGQ is sufficient to reduce communication traffic while maintaining good model accuracy, we recommend to use this as the default training configuration in practice.

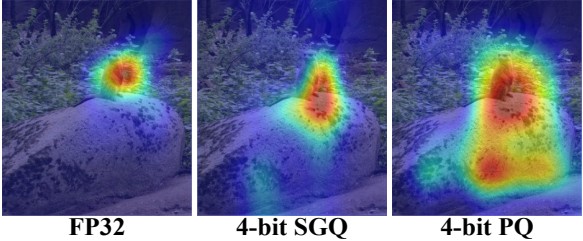

**FP32**          **4-bit SGQ**          **4-bit PQ**

Figure 7: The heat maps of the output features generated by different training schemes, which highlight the class-discriminative regions of model's prediction results.

**Visualization of quantized features.** Based on the training of ShuffleNet-1.0x with mini-ImageNet under different baselines, we employ the Grad-CAM++ [2] method to generate the heat maps of output features and inspect which feature regions impact the model's decision most. As shown in Figure 7, we can observe that SGQ holds a compact heat map as FP32 does, with similar texture and focal points. However, PQ's heat map skews far away from FP32's focal points with fuzzy texture. This indicates that SGQ can correctly localize significant regions while PQ may make mistakes, thus achieving a higher model accuracy over PQ.

### 4.5   System Overhead

Table 2: Average system overhead proportion (%) of computational time in different SGQ modules.

| # Bits | SGQ | CAG's FP | CAG's BP | GC | Total |
|---|---|---|---|---|---|
| 8 bits | 31.93 | 1.09 | 1.88 | 8.67 | 43.57 |
| 6 bits | 14.86 | 1.06 | 1.76 | 4.51 | 22.19 |
| **4 bits (Suggested)** | **5.51** | **1.05** | **1.75** | **1.62** | **9.89** |
| 2 bits | 4.12 | 1.04 | 1.73 | 1.24 | 8.14 |

Controlling system overhead is also a crucial issue to deploy SGQ on realistic edge devices. As shown in Table 2, we inspect the overhead proportion (%) of computational time cost in different modules during the training procedure. SGQ is the most fundamental module in the forward pass, where the vanilla K-means clustering dominates the computational time. By employing the pixel sampling based on average pooling, we could restrict the overhead proportion within $5.51\%$ when using the suggested 4-bit quantization. Meanwhile, by employing softmax-based approximation and affine transformation, the overhead of GC module is well bounded within $1.62\%$ during backward pass. Note that the CAG block involves both forward and backward passes as its FC layers requires parameter updating. CAG block holds slight overhead in these two stages and is independent to quantization bits, thus providing a good extensibility to general CNN models. Based on the three modules, the total overhead of SGQ is controlled in an acceptable range that matches the on-device computational capacities. In practice, we suggest using 4-bit quantization to make a balance between system overhead and model accuracy. In such setting, our SGQ method could improve the image processing speed (images/sec) and achieve good speedup from $9.22\times$ to $11.37\times$, on average, over the baselines.

## 5 Conclusion

This work develops new insights into traffic saving to build a communication-efficient collaborative learning paradigm. Unlike previous methods aiming at improving bandwidth utilization or using an unstructured pixel-wise compression, we jointly capture the channel and spatial-level feature redundancy, and conduct a hierarchical compression in these two levels to achieve a much higher traffic reduction ratio. Specifically, we propose the *Stripe-wise Group Quantization* (SGQ) method to better leverage the pixel similarity by reorganizing the features into groups based on channel significance, handled by the *Channel-attention Grouping* (CAG) block in forward pass. Meanwhile, we calibrate the gradients of quantized features with a comprehensive theoretical analysis of the convergence rate. Evaluations show that SGQ provides a significant traffic reduction over existing methods while not sacrificing much model accuracy under different quantization bits, achieving good training flexibility and communicational efficiency. We believe SGQ can contribute to the further development of edge intelligence applications.

## 6 Acknowledgements

This research was supported by fundings from the Key-Area Research and Development Program of Guangdong Province (No. 2021B0101400003), Hong Kong RGC Research Impact Fund (No. R5060-19), General Research Fund (No. 152221/19E, 152203/20E, and 152244/21E), the National Natural Science Foundation of China (61872310), and Shenzhen Science and Technology Innovation Commission (JCYJ20200109142008673).

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
