# A  Detailed Notations of SGQ in Sec. 2.1

Table 3: Notation list.

| Notation | Description |
|---|---|
| $N$ | The number of devices |
| $J$ | The cost function of K-means clustering |
| $Q(\cdot)$ | The pixel encoding function of SGQ |
| $\mathbf{X}$ | The collection of all pixels belonging to the features |
| $\boldsymbol{x}$ | The original full-precision pixel that $\boldsymbol{x}_i \in \mathbf{X}$ |
| $\hat{\boldsymbol{x}}$ | The quantized pixel that $\hat{\boldsymbol{x}}_i = Q(\boldsymbol{x}_i)$ |
| $\boldsymbol{u}$ | The clustering centroid hold the same dimension as pixel $\boldsymbol{x}$ |
| $\mathbf{U}$ | The matrix of all clustering centroids |
| $\mathbf{U}^+$ | The generalized inverse of $\mathbf{U}$ that follows $\mathbf{U}\mathbf{U}^+\mathbf{U} = \mathbf{U}$ |
| $\mathbf{Y}$ | The matrix of all cluster labels |
| $y_i$ | The cluster label corresponding to pixel $\boldsymbol{x}_i$ and $y_i \in \mathbf{Y}$ |
| $\mathbf{R}$ | The matrix of pixel mapping |
| $r_{i,j}$ | The $j$-th row of $\mathbf{R}$ indicating whether pixel $\boldsymbol{x}_i$ belongs to cluster label $y_j$ in the one-hot form |
| $C$ | The number of channels |
| $\frac{\partial Q(\boldsymbol{x})}{\partial \boldsymbol{x}}$ | The approximate gradients of the quantized features |
| $\boldsymbol{g}_x$ | The real gradients of the quantized features |
| $\boldsymbol{p}$ | The weight vector of clustering centroid |
| $f(\cdot)$ | The loss function |
| $\|\cdot\|_2$ | The $\ell_2-$norm |

All the notations used in the supplementary material are listed in Table 3.

# B  Details of Gradient Calibration in Sec. 2.3

Here, we present the detailed analysis of gradient calibration. As to the formulation of our SGQ method, we use softmax function to approximate the one-hot cluster label and the pixel encoding function is described as:

$$\hat{\boldsymbol{x}}_i = Q(\boldsymbol{x}_i) = \frac{\sum_{j=1}^{k} p_j \boldsymbol{u}_j}{\sum_{j=1}^{k} p_j}, \tag{16}$$

where $p_j = e^{-(\boldsymbol{x}_i - \boldsymbol{u}_j)(\boldsymbol{x}_i - \boldsymbol{u}_j)^\top}$. Therefore, pixel encoding can be regarded as a weighted mean of each clustering centroid. Let $\boldsymbol{p}$ be the weight vector of encoded pixels, we have:

$$\boldsymbol{p} = \begin{bmatrix} \frac{p_1}{\Sigma_{j=1}^k p_j} & \frac{p_2}{\Sigma_{j=1}^k p_j} & \cdots & \frac{p_k}{\Sigma_{j=1}^k p_j} \end{bmatrix}, \tag{17}$$

$$\hat{\boldsymbol{x}}_i = Q(\boldsymbol{x}_i) = \boldsymbol{p} \cdot \mathbf{U}, \tag{18}$$

$$\boldsymbol{p} = \hat{\boldsymbol{x}}_i \cdot \mathbf{U}^+, \tag{19}$$

where $\mathbf{U}^+$ is the generalized inverse of $\mathbf{U}$ that follows $\mathbf{U}\mathbf{U}^+\mathbf{U} = \mathbf{U}$. Then, the detailed calculation of the gradients based on quantized feature maps can be described as:

$$
\begin{aligned}
\frac{\partial Q(\boldsymbol{x})}{\partial \boldsymbol{x}} &= \frac{-2\sum_{j=1}^k p_j \boldsymbol{u}_j^\top(\boldsymbol{x}-\boldsymbol{u}_j)\sum_{j=1}^k p_j + 2\sum_{j=1}^k p_j \boldsymbol{u}_j^\top \sum_{j=1}^k p_j(\boldsymbol{x}-\boldsymbol{u}_j)}{(\sum_{j=1}^k p_j)^2}, \\
&= \frac{-2\sum_{j=1}^k \sum_{l=1}^k p_j p_l \boldsymbol{u}_j^\top(\boldsymbol{x}-\boldsymbol{u}_j) + 2\sum_{j=1}^k \sum_{l=1}^k p_j p_l \boldsymbol{u}_j^\top(\boldsymbol{x}-\boldsymbol{u}_l)}{\sum_{j=1}^k \sum_{l=1}^k p_j p_l}, \\
&= \frac{2\sum_{j=1}^k \sum_{l=1}^k p_j p_l \boldsymbol{u}_j^\top(\boldsymbol{u}_j - \boldsymbol{u}_l)}{\sum_{j=1}^k \sum_{l=1}^k p_j p_l}, \\
&= \frac{2\sum_{j=1}^k \sum_{l=1}^k p_j p_l \boldsymbol{u}_j^\top \boldsymbol{u}_j}{\sum_{j=1}^k \sum_{l=1}^k p_j p_l} - \frac{2\sum_{j=1}^k p_j \boldsymbol{u}_j^\top \sum_{l=1}^k p_l \boldsymbol{u}_l}{\sum_{j=1}^k p_j \sum_{l=1}^k p_l}, \\
&= \frac{2\sum_{j=1}^k p_j \boldsymbol{u}_j^\top \boldsymbol{u}_j}{\sum_{j=1}^k p_j} - 2\hat{\boldsymbol{x}}^\top \cdot \hat{\boldsymbol{x}}.
\end{aligned}
\tag{20, 21}
$$

As the polynomial term $\frac{2\sum_{j=1}^k p_j \boldsymbol{u}_j^\top \boldsymbol{u}_j}{\sum_{j=1}^k p_j}$ holds an analogous expression of pixel encoding function, we introduce a matrix $\mathbf{U}_o$ and use dot product to simplify the gradient formulation as follows.

$$\frac{\partial Q(\boldsymbol{x})}{\partial \boldsymbol{x}} = 2\hat{\boldsymbol{x}} \cdot \mathbf{U}^+ \cdot \mathbf{U}_o - 2\hat{\boldsymbol{x}}^\top \cdot \hat{\boldsymbol{x}}, \tag{22}$$

$$\mathbf{U}_o = \begin{bmatrix} \boldsymbol{u}_1^\top \boldsymbol{u}_1 \\ \boldsymbol{u}_2^\top \boldsymbol{u}_2 \\ \cdots \\ \boldsymbol{u}_k^\top \boldsymbol{u}_k \end{bmatrix}, \tag{23}$$

$$\frac{\sum_{j=1}^k p_j \boldsymbol{u}_j^\top \boldsymbol{u}_j}{\sum_{j=1}^k p_j} = \boldsymbol{p} \cdot \mathbf{U}_o. \tag{24}$$

## C Details of Convergence Analysis and Theorem Proof in Sec. 3

**Theorem 1.** *Assuming each pixel follows $\|\boldsymbol{x}\|_2 \leq B$, the upper bound of the approximate gradients can be described as $\|\frac{\partial Q(\boldsymbol{x})}{\partial \boldsymbol{x}}\|_2 \leq 2B^2$. Meanwhile, given $m$ pixels and $C$ channels, the upper bound of real gradients is described as $\frac{C}{m} \leq \|\boldsymbol{g}_x\|_2 \leq C$. Therefore, the approximate gradients calculated by GC follows $\|\frac{\partial Q(\boldsymbol{x})}{\partial \boldsymbol{x}}\|_2 \leq \frac{2B^2 m}{C}\|\boldsymbol{g}_x\|_2$.*

*Proof.* Considering the fact that cluster centroid $\boldsymbol{u}_j$ holds the same upper bound as pixel $\boldsymbol{x}$, we can obtain the upper bound of $\boldsymbol{u}_j$ as:

$$
\begin{aligned}
\|\boldsymbol{u}_j\|_2 &= \|\frac{\sum_{i=1}^m r_{i,j} \boldsymbol{x}_i}{\sum_{i=1}^m r_{i,j}}\|_2 \\
&\leq \sum_{i=1}^m \|\frac{r_{i,j} \boldsymbol{x}_i}{\sum_{i=1}^m r_{i,j}}\|_2, \\
&\leq \sum_{i=1}^m \frac{r_{i,j} B}{\sum_{i=1}^m r_{i,j}} = B.
\end{aligned}
\tag{25}
$$

Then, we present the bound of the real gradient $\boldsymbol{g}_x$ generated by the SGQ method. The pixel encoding function $Q(\cdot)$ partitions the entire feature maps into several clusters with unique labels and replaces each pixel $\boldsymbol{x}$ by the corresponding clustering centroid. Therefore, the gist of SGQ is to map the original pixel $\boldsymbol{x}$ to the clustering centroid $\boldsymbol{u}_j$ and the updating process of cluster centroid $\boldsymbol{u}_j$ is controlled by the following function:

$$\boldsymbol{u_j} = \frac{\sum_{i=1}^{m} r_{i,j} \boldsymbol{x_i}}{\sum_{i=1}^{m} r_{i,j}}, \tag{26}$$

where $1 \leq \sum_{i=1}^{m} r_{i,j} \leq m$. Thus we can figure out the bound as:

$$\frac{C}{m} \leq \|\boldsymbol{g}_x\|_2 \leq C. \tag{27}$$

Based on the the gradients of quantized feature maps in Eq. (20), we can deduce the gradient bound of $Q(\boldsymbol{x})$ as:

$$
\begin{aligned}
\|\frac{\partial Q(\boldsymbol{x})}{\partial \boldsymbol{x}}\|_2 &= \|\frac{2\sum_{j=1}^{k}\sum_{l=1}^{k} \boldsymbol{u}_j^\top (\boldsymbol{u}_j - \boldsymbol{u}_l) p_j p_l}{\sum_{j=1}^{k}\sum_{l=1}^{k} p_j p_l}\|_2, \\
&\leq 2\sum_{j=1}^{k}\sum_{l=1}^{k} \|\frac{\boldsymbol{u}_j^\top (\boldsymbol{u}_j - \boldsymbol{u}_l) p_j p_l}{\sum_{j=1}^{k}\sum_{l=1}^{k} p_j p_l}\|_2, \\
&\leq 2\sum_{j=1}^{k}\sum_{l=1}^{k} \frac{\|\boldsymbol{u}_j^\top\|_2 \|\boldsymbol{u}_j - \boldsymbol{u}_l\|_2 p_j p_l}{\sum_{j=1}^{k}\sum_{l=1}^{k} p_j p_l}, \\
&\leq 2\sum_{j=1}^{k}\sum_{l=1}^{k} \frac{B^2 p_j p_l}{\sum_{j=1}^{k}\sum_{l=1}^{k} p_j p_l}, \\
&= 2B^2. \tag{28}
\end{aligned}
$$

Therefore, the upper bound of approximate and real gradients can be described as follows, respectively:

$$\|\frac{\partial Q(\boldsymbol{x})}{\partial \boldsymbol{x}}\|_2 \leq 2B^2, \tag{29}$$

$$\frac{C}{m} \leq \|\boldsymbol{g}_x\|_2 \leq C. \tag{30}$$

Solving $\|\frac{\partial Q(\boldsymbol{x})}{\partial \boldsymbol{x}}\|_2 \leq a\|\boldsymbol{g}_x\|_2$ yields $0 < a \leq \frac{2B^2 m}{C}$, which is easy to hold by the feature maps of common models.

Given the loss function $f(\cdot)$ and corresponding gradients $\nabla f(\boldsymbol{w}_t)$, the parameter updating rule under SGD optimizer can be formulated as:

$$\boldsymbol{w}_{t+1} = \boldsymbol{w}_t - \gamma \frac{1}{N} \sum_{i=1}^{N} Q[g_i(\boldsymbol{w}_t)], \tag{31}$$

where $\gamma = \eta a$ and $\eta$ represents the learning rate, $N$ is the number of devices, and $Q[g_i(\boldsymbol{w}_t)]$ is the local gradient computed by node $i$ under feature quantization.

Based on the assumption of Lipschitz-continuous objective gradients such that $\|\nabla F(\omega_1) - \nabla F(\omega_2)\|_2 \leq L\|\omega_1 - \omega_2\|_2$ for all $\omega_1, \omega_2$, where $L$ is the Lipschitz constant, we have:

$$Ef(\boldsymbol{w}_{t+1}) - Ef(\boldsymbol{w}_t)$$

$$\leq \quad E\langle \boldsymbol{w}_{t+1} - \boldsymbol{w}_t, \nabla f(\boldsymbol{w}_t)\rangle + \frac{L}{2}E\|\boldsymbol{w}_{t+1} - \boldsymbol{w}_t\|_2^2$$

$$= \quad -\gamma E\langle \frac{1}{N}\sum_{i=1}^{N}Q[g_i(\boldsymbol{w}_t)], \nabla f(\boldsymbol{w}_t)\rangle + \frac{L\gamma^2}{2}E\|\frac{1}{N}\sum_{i=1}^{N}Q[g_i(\boldsymbol{w}_t)]\|_2^2$$

$$= \quad -\frac{\gamma}{2}\|\nabla f(\boldsymbol{w}_t)\|_2^2 - \frac{\gamma}{2}\|\frac{1}{N}\sum_{i=1}^{N}Q[g_i(\boldsymbol{w}_t)]\|_2^2 + \frac{\gamma}{2}\|\nabla f(\boldsymbol{w}_t) - \frac{1}{N}\sum_{i=1}^{N}Q[g_i(\boldsymbol{w}_t)]\|_2^2$$

$$+ \frac{L\gamma^2}{2N^2}E\|\sum_{i=1}^{N}Q[g_i(\boldsymbol{w}_t)] - \sum_{i=1}^{N}g_i(\boldsymbol{w}_t) + \sum_{i=1}^{N}g_i(\boldsymbol{w}_t) - N\nabla f(\boldsymbol{w}_t) + N\nabla f(\boldsymbol{w}_t)\|_2^2$$

$$\leq \quad (\frac{3L\gamma^2}{2} - \gamma)E\|\nabla f(\boldsymbol{w}_t)\|_2^2 + \frac{3L\gamma^2\delta^2 B^2 m}{NC}, \tag{32}$$

where the second equality comes from $<a, b> = \frac{1}{2}\|a\|^2 + \frac{1}{2}\|b\|^2 - \frac{1}{2}\|a-b\|^2$, and the last inquality holds according the the widely used variance assumption in SGD such that $\|g_i(\boldsymbol{w}_t) - \nabla F(\boldsymbol{w}_t)\| \leq \delta^2$ ($\delta$ is a constant).

By accumulating both sides of Eq. (32) from $t = 0$ to $t = T - 1$ and dividing the both side by $\gamma T$, we obtain:

$$(1 - \frac{L\gamma}{2})\frac{1}{T}\sum_{t=0}^{T-1}E\|\nabla f(\boldsymbol{w}_t)\|_2^2 \leq \frac{Ef(\boldsymbol{w}_0) - Ef(\boldsymbol{w}^*)}{\gamma T} + \frac{3L\gamma\delta^2 B^2 m}{NCT} \tag{33}$$

where $f(\boldsymbol{w}^*)$ is the optimal solution of parameter updating. Let $\eta = O(\frac{\sqrt{N}}{aL\sqrt{T}})$, then for sufficiently large $T$, we have:

$$\frac{1}{T}\sum_{t=0}^{T-1}E\|\nabla f(\boldsymbol{w}_t)\|_2^2 \preceq O(\frac{1}{\sqrt{NT}}), \tag{34}$$

where $\preceq$ denotes order inequality, which means less than or equal to up to a constant factor. Consequently, the proposed SGQ holds the same order of convergence rate as the non-quantized distributed SGD algorithm and exhibits the linear speedup property with respect to the number of devices. The theoretical results demonstrate that the proposed algorithm is communication-efficient and scalable.

$\square$

## D   Details of Convergence Efficiency using Different Bits in Sec. 4.4

As shown in Figure 8, we present the detailed training convergence curves by using SGQ under different quantization bits and benchmarks. SGQ achieves acceptable accuracy in most cases, even for the 3-bit training (*e.g.,* ResNet34 and ShuffleNet-1.0x on mini-ImageNet). These results verify that the proposed SGQ method can effectively compress feature size without sacrificing much model accuracy as FP32 training. In realistic deployment, as our SGQ method under 4 bits is sufficient to fundamentally reduce communication traffic while maintaining good model accuracy, we recommend to use the 4-bit SGQ as the default training configuration.

## E   Further Analysis of Large-scale Performance

Considering the objective of establishing communication-efficient collaborative learning on edge devices, the evaluations follow the typical experimental setting adopted by previous works [39, 12], including the configurations of comparison baselines, datasets, neural network models, deployment devices, and bandwidth environment.

Note that the edge devices usually hold much lower computational and storage capacity over the conventional GPU-based machines in the cloud. Such a resource-constrained environment limits

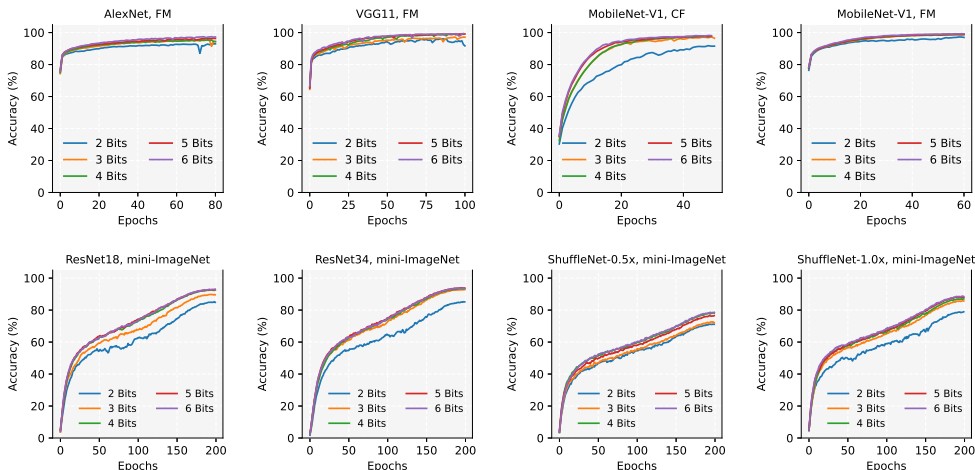

Figure 8: Details of training convergence curves under different quantization bits.

Table 4: Comparison of average model accuracy (%) using 8-bit compression.

| Method | ResNet50 | ResNeXt101 | MobileNet-V3 | EfficientNet |
|---|---|---|---|---|
| FP32 (Acc. Upper Bound) | 75.56 | 77.17 | 72.13 | 81.47 |
| UQ | 69.34 | 70.68 | 54.93 | 61.86 |
| PQ | 72.25 | 72.99 | 60.84 | 67.59 |
| Top-k CLIO | 11.89 | 12.46 | 10.18 | 13.23 |
| **SGQ** | 74.67 | 75.43 | 70.39 | 80.15 |

Table 5: Comparison of traffic size (MB).

| Method | ResNet50 | ResNeXt101 | MobileNet-V3 | EfficientNet |
|---|---|---|---|---|
| FP32 | 143.11 | 169.63 | 108.61 | 85.22 |
| UP | 36.60 | 43.38 | 27.78 | 21.80 |
| PQ | 26.35 | 33.32 | 18.37 | 17.85 |
| Top-k CLIO | 55.50 | 66.24 | 33.33 | 37.03 |
| SGQ | 10.54 | 13.33 | 8.82 | 7.25 |

Table 6: Comparison of inference speed (ms).

| Method | ResNet50 | ResNeXt101 | MobileNet-V3 | EfficientNet |
|---|---|---|---|---|
| FP32 | 124.10 | 147.10 | 94.19 | 73.90 |
| UP | 40.89 | 48.47 | 31.04 | 24.35 |
| PQ | 32.89 | 40.61 | 23.69 | 21.27 |
| Top-k CLIO | 55.66 | 66.33 | 35.37 | 36.25 |
| SGQ | 20.54 | 24.99 | 16.22 | 12.99 |

Table 7: Comparison of processing throughput (images/sec).

| Method | ResNet50 | ResNeXt101 | MobileNet-V3 | EfficientNet |
|---|---|---|---|---|
| FP32 | 8 | 7 | 11 | 14 |
| UP | 24 | 21 | 32 | 41 |
| PQ | 30 | 25 | 42 | 47 |
| Top-k CLIO | 18 | 15 | 28 | 28 |
| SGQ | 49 | 40 | 62 | 77 |

the choice of evaluation datasets. It is impractical if we directly deploy the original ImageNet-1K (1.2 million images with more than 140 GB storage demands) dataset on the commodity edge device (*e.g.,* the typical NVIDIA Jetson Nano with 4GB RAM and 64 GB micro SD storage). Actually, the mini-ImageNet is a 100-class subset of ImageNet for one-shot learning and is more suitable for the evaluation based on the tiny edge devices. In real-world scenarios, edge devices often hold quite limited labeled data but need to handle the edge intelligence applications.

Actually, our method can apply to large-scale tasks if we are equipped with powerful machines. We verify this point by conducting the experiments on the ImageNet-1K datasets with ResNet50, ResNeXt101, MobileNetV2 and EfficientNet. As to the hardware configuration, We use a machine with NVIDIA 3090 GPU to replace the original edge device with the tiny Jetson Nano board, so as to provide sufficient computational and storage capacity for handling ImageNet. Other experimental configurations are the same as the main paper. Under the unified setting of using 8-bit compression and mini-batch training, we observe that SGQ consistently outperforms all the baselines, on average, in terms of model accuracy (%) in Table 4, traffic size (MB) in Table 5, inference speed (ms) in Table 6 and image processing throughput (images/sec) in Table 7. These four tables present a comprehensive inspection on large-scale performance. We can observe that SGQ effectively outperforms the other methods in different metrics and achieves a good quality-traffic trade-off.