# OpenReview forum: "Hierarchical Channel-spatial Encoding for Communication-efficient Collaborative Learning"
_NeurIPS.cc/2022/Conference — NeurIPS 2022 Accept_

### Official Review · Reviewer_HVqg · 2022-07-07

**Rating:** 6
**Confidence:** 3
**Soundness:** 3 good
**Presentation:** 3 good
**Contribution:** 3 good

**Summary:**

This work proposes a structured strip-wise quantization method, named Strip-Wise Group Quantization (SGQ), to implement the scalable collaborative learning systems on tiny edge devices through a hierarchical compression on refactored feature structures.

**Questions:**

1. Dose K-means used for dividing the feature maps into several clusters take a long time? As per my understanding, K-means tend to be way slow for grouping, and the cluster number is also hand-tuned. Now that the Channel-Attention Grouping block is employed to measure and group the entire features, I was wondering if attention could further support dividing them into smaller groups based on pixel similarity instead of using K-means.
2. Is it possible to employ extra attention, like spatial attention for spatial dimension, as the channel-attention block for channel dimension in this work? If not, why could we do attention to the channel dimension but not the spatial dimension?

**Strengths And Weaknesses:**

Strengths
1. The novelty of capturing such channel-dimension structured information to compress feature size is good, which is different from the viewpoint of the conventional methods.
2. The authors deliver a clear set of challenges and motivations for deploying CL systems on tiny edge devices.

Weaknesses
1. This article addresses the communication bottleneck and builds a novel communication-efficient CL system; it is better to bring out a well-through analysis in the experiment section. There are not enough demonstrations regarding communication efficiency.
2. The analysis of the sublinear convergence order of SGQ is not clear in Section 3.
3. The ablation study is based on the performance of the final model with different quantization bits, which shows the impact of quantization bits. I think it would be better if the performance of other components of SGQ could be added to analyze their different effects on the final model.

---

> ### Author Response · Authors · 2022-08-01
> **Author Response to Reviewer HVqg (Part II)**
>
> **Q3:** Additional response to the convergence order of SGQ in Section 3.
> **A3:** Due to the page limit, we present the Theorem in Section 3 of the main submission and give detailed proof in the supplementary material in Section C. The proposed SGQ holds **the same order of convergence rate as the non-quantized distributed SGD algorithm** and exhibits the **linear speedup property with respect to the number of devices**. The theoretical results demonstrate that our proposed algorithm is communication-efficient and scalable.
>
> **Q4:** Additional response to the ablation study of SGQ's different components.
> **A4:** Additionally, we also inspect how each component of SGQ impacts the model accuracy, compared with the original FP32 training (theoretical upper bound). The ablation study contains the components of Quantization, Channel-attention Grouping (CAG), and Gradient Calibration (GC). As the 4-bit SGQ is sufficient to reduce communication traffic while maintaining good model accuracy, we recommend using this as the default training configuration in practice. We present the results in the following table.
>
> **Table: Ablation study of how each component impacts the model accuracy.**
> | Dataset, Model                     |                 Configuration | Top-1 Acc. (%) | Drop w.r.t FP32 (%) |
> |--------------------------------------------|------------------------------:|---------------:|--------------------:|
> | Fashion MNIST, VGG-11              |            FP32 (Acc. Upper Bound) |          97.55 |                   - |
> | Fashion MNIST, VGG-11              |       SGQ (Quantization Only) |          92.91 |                4.64 |
> | Fashion MNIST, VGG-11              |        SGQ (Quantization+CAG) |          95.41 |                2.14 |
> | **Fashion MNIST, VGG-11**          | **SGQ (Quantization+CAG+GC)** |      **96.57** |            **0.98** |
> | CIFAR-10, MobileNet-V1             |            FP32 (Acc. Upper Bound) |          94.74 |                   - |
> | CIFAR-10, MobileNet-V1             |       SGQ (Quantization Only) |          91.39 |                3.35 |
> | CIFAR-10, MobileNet-V1             |        SGQ (Quantization+CAG) |          93.02 |                1.72 |
> | **CIFAR-10, MobileNet-V1**         | **SGQ (Quantization+CAG+GC)** |      **93.45** |            **1.29** |
> | CIFAR-100, ResNet-34                |            FP32 (Acc. Upper Bound) |          72.19 |                   - |
> | CIFAR-100, ResNet-34               |       SGQ (Quantization Only) |           56.5 |               15.69 |
> | CIFAR-100, ResNet-34               |        SGQ (Quantization+CAG) |          68.63 |                3.56 |
> | **CIFAR-100, ResNet-34**           | **SGQ (Quantization+CAG+GC)** |      **70.64** |            **1.55** |
> | mini-ImageNet, ShuffleNet-1.0x     |            FP32 (Acc. Upper Bound) |          78.73 |                   - |
> | mini-ImageNet, ShuffleNet-1.0x     |       SGQ (Quantization Only) |          70.01 |                8.72 |
> | mini-ImageNet, ShuffleNet-1.0x     |        SGQ (Quantization+CAG) |          73.88 |                4.85 |
> | **mini-ImageNet, ShuffleNet-1.0x** | **SGQ (Quantization+CAG+GC)** |      **74.86** |            **3.87** |
>
> From the above table, we can observe that simply introducing the stripe-wise quantization without CAG will cause a significant accuracy drop compared with the FP32 training, which is unacceptable in realistic scenarios. This indicates the significance of feature grouping based on channel attention.
> In contrast, the model accuracy is preserved by employing the CAG block and GC method, where CAG  extracts the most informative features to improve quantization efficiency and GC adjusts the gradients of quantized features to achieve a stable convergence rate.
> Therefore, the CAG block and GC method are the cores to guarantee SGQ's training efficiency.

---

> ### Author Response · Authors · 2022-08-01
> **Author Response to Reviewer HVqg (Part I)**
>
> Thank you very much for your valuable comments and recognition of our work. We will reply to each question as follows.
>
> **Q1:** Dose K-means used for dividing the feature maps into several clusters take a long time? As per my understanding, K-means tend to be way slow for grouping, and the cluster number is also hand-tuned. Now that the Channel-Attention Grouping block is employed to measure and group the entire features, I was wondering if attention could further support dividing them into smaller groups based on pixel similarity instead of using K-means.
> **A1:** Restricting the computational cost is important to deploy SGQ on realistic edge devices. The K-means clustering is used in the  feature discretization stage inside SGQ, where we carefully take the property of K-means clustering into consideration and well control its overhead. More precisely, we adopt the K-means clustering inside each channel group (referred to the Eq.8 in the main submission) to obtain the cluster centroid for pixel encoding. We downsample the feature maps first and conduct K-means clustering based on the downscaled samples, instead of the whole feature map. Meanwhile, as we conduct the stripe-wise quantization along the channel dimension, the clustering space follows the scale of $O(W \times H)$, where $W$ and $H$ represent the width and height of the pixel plane, respectively. As mentioned by the case study in Eq.10,  $O(W \times H)$ holds a relatively smaller scale compared with the original feature map with a scale of $O(W \times H \times \sum_{i=1}^G C_i)$.
> Therefore, our K-means clustering module does not take a long time. Instead, its time cost proportion is **less than 6%** during inference (or the forward pass) when using the 4-bit SGQ. The detailed computation costs of each component are summarized in Table 2 in the main submission and we present the results here. Note that the K-means clustering cost is counted in the column of SGQ.
>
> **Table: Average system overhead proportion (%) of computational time in different SGQ modules.**
> | # Bits                 |      SGQ | CAG's Forward Pass | CAG's Backward Pass | Gradient Calibration (GC) |    Total |
> |------------------------|---------:|-------------------:|--------------------:|--------------------------:|---------:|
> | 8 Bits                 |    31.93 |               1.09 |                1.88 |                      8.67 |    43.57 |
> | 6 Bits                 |    14.86 |               1.06 |                1.76 |                      4.51 |    22.19 |
> | **4 Bits (Suggested)** | **5.51** |           **1.05** |            **1.75** |                  **1.62** | **9.89** |
> | 2 Bits                 |     4.12 |               1.04 |                1.73 |                      1.24 |     8.14 |
>
> **Q2:** Is it possible to employ extra attention, like spatial attention for spatial dimension, as the channel-attention block for channel dimension in this work? If not, why could we do attention to the channel dimension but not the spatial dimension?
> **A2:** It is a valuable inspiration to further study the opportunity that employing the CAG-like module to replace the K-means clustering for pixel grouping in the spatial dimension. The key here is to precisely determine each pixel's significance or the representation semantics for the task (e.g., feature attention to classify an image) within the entire feature map. Then, we can group pixels with close significance together. The major reason why we employ attention to the channel dimension instead of the spatial dimension is the computational cost of the inside fully-connected layers for calculating the element (i.e., channels for channel attention and pixels for spatial attention, respectively) significance. As the number of channels inside each group is much fewer than pixels ($C \ll W \times H$), conducting channel attention is computationally affordable on the edge devices. However, we believe that optimizing spatial attention with less overhead will be an interesting problem for further study.

---

### Official Review · Reviewer_B9sp · 2022-07-10

**Rating:** 5
**Confidence:** 1
**Soundness:** 3 good
**Presentation:** 2 fair
**Contribution:** 2 fair

**Summary:**

Network bandwidth poses challenges to the collaborative learning system. The existing communication-optimized methods either are insufficient in reducing the traffic or may degrade the model performance. This work proposed a hierarchical compression algorithm for feature data, called Stripe-wise Group Quantization (SGQ). SGQ refactors feature based on similarity in both channel and spatial levels. This work also presented the theoretical analysis for the convergence order of SGQ. Experiments show that SGQ offers a higher reduction ratio without accuracy loss compared to state-of-the-art methods such as Product Quantization (PQ).

**Questions:**

Please refer to the above weakness part.

**Limitations:**

The authors have adequately addressed the limitations and potential negative societal impact of their work.

**Strengths And Weaknesses:**

Strengths:
+ This work tried to tackle the communication challenges of the collaborative learning system deployed in reality.  Experiments show that the proposed method SGQ is very practical:
   + SGQ offers a sufficient reduction ratio in the data traffic
   + SGQ maintains the model accuracy and the convergence is proved
   + the overhead of SGQ is only around 8-9% of the total runtime, which is acceptable compared to the communication overhead, and thus SGQ is able to offer 9.22X to 11.37X speedup.
+ The paper is well organized and easy to follow.

Weaknesses:
- The evaluation section only shows the traffic reduction ratio and traffic size. It would be better to show the tradeoff between accuracy and actual measured latency and throughput.
- It is unclear the contribution of each part of SGQ. It would be better to have the ablation study on different parts of SGQ.

---

> ### Author Response · Authors · 2022-08-01
> **Author Response to Reviewer B9sp (Part II)**
>
> **Q2:** It is unclear the contribution of each part of SGQ. It would be better to have the ablation study on different parts of SGQ.
> **A2:** Yes, it is important to inspect the contribution of each part of SGQ. Here, we intend to present the alation studies in terms of (1) computational cost and (2) model accuracy.
> **(1) Computational Cost.** In our main submission, we discuss the computational cost (i.e., the average time proportion %) of each component of SGQ in Table 2. The results show that the overhead of each part of SGQ is controlled in an acceptable range that matches the on-device computational capacities. In such a setting, our SGQ method could improve the image processing speed (images/sec) and achieve a good speedup from 9.22× to 11.37×, on average, over the baselines. The results are listed in the following table.
>
> **Table: Average system overhead proportion (%) of computational time in different SGQ modules.**
> | # Bits                 |      SGQ | CAG's Forward Pass | CAG's Backward Pass | Gradient Calibration (GC) |    Total |
> |------------------------|---------:|-------------------:|--------------------:|--------------------------:|---------:|
> | 8 Bits                 |    31.93 |               1.09 |                1.88 |                      8.67 |    43.57 |
> | 6 Bits                 |    14.86 |               1.06 |                1.76 |                      4.51 |    22.19 |
> | **4 Bits (Suggested)** | **5.51** |           **1.05** |            **1.75** |                  **1.62** | **9.89** |
> | 2 Bits                 |     4.12 |               1.04 |                1.73 |                      1.24 |     8.14 |
>
> **(2) Model Accuracy.** Additionally, we also inspect how much improvement of model accuracy is achieved by each part of SGQ, i.e., Quantization, Channel-attention Grouping (CAG), and Gradient Calibration (GC). As the 4-bit SGQ is sufficient to reduce communication traffic while maintaining good model accuracy, we recommend using this as the default training configuration in practice. We present the results in the following table.
>
> **Table: Ablation study of how each component impacts the model accuracy.**
> | Dataset, Model                     |                 Configuration | Top-1 Acc. (%) | Drop w.r.t FP32 (%) |
> |--------------------------------------------|------------------------------:|---------------:|--------------------:|
> | Fashion MNIST, VGG-11              |            FP32 (Acc. Upper Bound) |          97.55 |                   - |
> | Fashion MNIST, VGG-11              |       SGQ (Quantization Only) |          92.91 |                4.64 |
> | Fashion MNIST, VGG-11              |        SGQ (Quantization+CAG) |          95.41 |                2.14 |
> | **Fashion MNIST, VGG-11**          | **SGQ (Quantization+CAG+GC)** |      **96.57** |            **0.98** |
> | CIFAR-10, MobileNet-V1             |            FP32 (Acc. Upper Bound) |          94.74 |                   - |
> | CIFAR-10, MobileNet-V1             |       SGQ (Quantization Only) |          91.39 |                3.35 |
> | CIFAR-10, MobileNet-V1             |        SGQ (Quantization+CAG) |          93.02 |                1.72 |
> | **CIFAR-10, MobileNet-V1**         | **SGQ (Quantization+CAG+GC)** |      **93.45** |            **1.29** |
> | CIFAR-100, ResNet-34                |            FP32 (Acc. Upper Bound) |          72.19 |                   - |
> | CIFAR-100, ResNet-34               |       SGQ (Quantization Only) |           56.5 |               15.69 |
> | CIFAR-100, ResNet-34               |        SGQ (Quantization+CAG) |          68.63 |                3.56 |
> | **CIFAR-100, ResNet-34**           | **SGQ (Quantization+CAG+GC)** |      **70.64** |            **1.55** |
> | mini-ImageNet, ShuffleNet-1.0x     |            FP32 (Acc. Upper Bound) |          78.73 |                   - |
> | mini-ImageNet, ShuffleNet-1.0x     |       SGQ (Quantization Only) |          70.01 |                8.72 |
> | mini-ImageNet, ShuffleNet-1.0x     |        SGQ (Quantization+CAG) |          73.88 |                4.85 |
> | **mini-ImageNet, ShuffleNet-1.0x** | **SGQ (Quantization+CAG+GC)** |      **74.86** |            **3.87** |
>
> From the above table, we can observe that simply introducing the stripe-wise quantization without CAG will cause a significant accuracy drop compared with the FP32 training, which is unacceptable in realistic scenarios. This indicates the significance of feature grouping based on channel attention.
> In contrast, the model accuracy is preserved by employing the CAG block and GC method, where CAG  extracts the most informative features to improve quantization efficiency and GC adjusts the gradients of quantized features to achieve a stable convergence rate.
> Therefore, the CAG block and GC method are the cores to guarantee SGQ's training efficiency.

---

> ### Author Response · Authors · 2022-08-01
> **Author Response to Reviewer B9sp (Part I)**
>
> Thank you very much for your constructive comments and recognition of our work. We will reply to each question as follows.
>
> **Q1:** The evaluation section only shows the traffic reduction ratio and traffic size. It would be better to show the tradeoff between accuracy and actual measured latency and throughput.
> **A1:** We are sorry for not presenting this part more clearly. In our main submission, we have discussed the trade-off between model accuracy and traffic size in Figure 5, where our SGQ explicitly outperforms existing methods. Note that the traffic size is directly related to the time cost for transmitting the intermediate features from the client to the server, thus impacting the entire inference latency and image processing throughput in the forward pass. Under the unified setting using the suggested 4-bit compression, we reorganize the experiments and present the trade-off in terms of average **top-1 accuracy (%)**, **latency (ms)** and **throughput (images/sec)** by adjusting the CNN architecture scales (e.g., tiny, small, compacted, medium and large). All the experimental configurations are consistent with our main submission. The results are summarized in the following table.
>
> **Table: Comparison of the tradeoff between accuracy, latency and throughput.**
> | **SGQ/Model Scale**  | **Top-1 Acc. (%)** | **Lantency (ms)** | **Throughput (images/sec)** |
> |----------|-------------------:|------------------:|----------------------------:|
> | Tiny  |              66.23 |             7.17  |                        139  |
> | Small  |              73.11 |            10.77  |                         93  |
> | Compacted  |              74.86 |            11.96  |                         84  |
> | Medium  |              76.89 |            12.56  |                         80  |
> | Large  |              77.67 |            14.36  |                         70  |
> | **PQ**   | **Top-1 Acc. (%)** | **Lantency (ms)** | **Throughput (images/sec)** |
> | Tiny   |              61.34 |            17.65  |                         57  |
> | Small   |              68.23 |            22.43  |                         45  |
> | Compacted   |              70.16 |            26.91  |                         37  |
> | Medium   |             72.435 |            31.40  |                         32  |
> | Large   |              73.97 |            35.89  |                         28  |
> | **UQ**   | **Top-1 Acc. (%)** | **Lantency (ms)** | **Throughput (images/sec)** |
> | Tiny   |             49.234 |            21.55  |                         46  |
> | Small   |              51.02 |            33.91  |                         29  |
> | Compacted   |              53.15 |            37.72  |                         27  |
> | Medium   |              55.32 |            43.61  |                         23  |
> | Large   |              60.23 |            46.73  |                         21  |
> | **CLIO** | **Top-1 Acc. (%)** | **Lantency (ms)** | **Throughput (images/sec)** |
> | Tiny |                9.7 |            36.62  |                         27  |
> | Small |                9.9 |            40.70  |                         25  |
> | Compacted |              11.1 |            44.20  |                         23  |
> | Medium |               11.9 |            66.13  |                         15  |
> | Large |              14.23 |            71.36  |                         14  |
>
> We can observe that our SGQ significantly outperforms existing methods in these trade-off experiments. Under different settings, SGQ can stably achieve a much higher top-1 accuracy over other methods, while still providing lower inference latency and higher image processing throughput.

---

> ### Author Response · Authors · 2022-08-09
> **Look Forward to Post-rebuttal**
>
> Thank you very much for spending time providing constructive and valuable comments on our paper. We have carefully replied to your questions and presented the supplementary experiments in the response.
>
> The experimental results verify that the proposed SGQ method can significantly outperform the existing methods in terms of **model accuracy (%), traffic size (MB), inference speed (ms) and image processing throughput (images/sec)**, even using the large-scale ImageNet-1K dataset. We also provided more detailed **alation studies** to inspect how each part of SGQ impacts the model accuracy, and compare the **system overhead** proportion (%) of computational time on each part. The evaluation shows that SGQ consistently achieves a **great tradeoff between accuracy, latency and throughput** under different model scales.
>
> Since the discussion period is quickly drawing to a close, we would like to know if there are any additional clarifications or experiments that we can offer. We are looking forward to your feedback and sincerely invite you to update the score if concerns are addressed.
>
> Thanks a lot for your time again!

---

### Official Review · Reviewer_ofcH · 2022-07-10

**Rating:** 6
**Confidence:** 3
**Soundness:** 3 good
**Presentation:** 4 excellent
**Contribution:** 3 good

**Summary:**

In order to improve the communication efficiency in collaborative learning systems, this paper proposes to consider not only the pixel similarities but also channel information by channel-attention grouping. Theoretical analysis is provided to calibrate gradients of both features and CAG blocks in the backward process. Experiments demonstrate higher traffic reduction and image processing speed while keeping comparable accuracy.

**Questions:**

1. What is the definition of $g_x$ in Theorem 1?
2. Since additional fully connected layers are added to learn the channel information, how about the increased computation cost when the number of channels and devices is high?


**Limitations:**

The number of clusters $k$ is tied to the bit, then the performance degradation of extreme cases including 1-bit or 2-bit is still a problem.

**Strengths And Weaknesses:**

Strengths:
1. The observation this work is based on is that some channels hold quite different features when the corresponding filters are orthogonal to each other. This phenomenon is quite interesting and makes it reasonable to design the channel-attention grouping block.
2. The analysis of gradients in the backward process is thorough and sound.
Experiments on NVIDIA Jetson Nano and HUAWEI Atlas 200DK demonstrate high efficiency and practical values of the proposed method with comparable performance with the baseline.
Weaknesses:
From the first few sections and figure 2, we would consider channel-attention grouping is in front of the stripe-wise group quantization, but these two methods are arranged in the opposite order in the methodology section. It seems to be confusing to readers.

---

> ### Author Response · Authors · 2022-07-31
> **Author Response to Reviewer ofcH**
>
> Thank you very much for your constructive comments and recognition of our work. We will reply to each question as follows.
>
> **Q1:** What is the definition of  $g_x$ in Theorem 1?
> **A1:** We denote $g_x$ as the real gradients of the quantized features. We have listed all the notations in Table 1 in the supplementary material. Here, we present them as follows for reading convenience.
>
> **Table: Notations used in the theoretical analysis.**
> | Notation | Description |
> |--------------|-------|
> |$N$ | The number of devices|
> |$J$ | The cost function of K-means clustering |
> |$Q(\cdot)$ | The pixel encoding function of SGQ |
> |$\mathbf{X}$ | The collection of all pixels belonging to the features |
> |$x$ | The original full-precision pixel that $x_i \in \mathbf{X}$ |
> |$\hat{x}$ | The quantized pixel that $\hat{x}_i = Q(x_i)$ |
> |$u$ | The clustering centroid holding the same dimension as pixel $x$ |
> |$\mathbf{U}$ | The matrix of all clustering centroids |
> |$\mathbf{U}^+$ | The generalized inverse of $\mathbf{U}$ that follows $\mathbf{U}\mathbf{U}^+\mathbf{U} = \mathbf{U}$|
> |$\mathbf{Y}$ | The matrix of all cluster labels |
> |$y_i$ | The cluster label corresponding to pixel $x_i$ and $y_i\in \mathbf{Y}$|
> |$\mathbf{R}$ |  The matrix of pixel mapping |
> |$r_{i,j}$ | The $j$-th row of $\mathbf{R}$ indicating whether pixel $x_i$ belongs to cluster label $y_j$  in the one-hot form |
> |$C$ | The number of channels|
> |$\frac{\partial Q(x)}{\partial x}$ | The approximate gradients of the quantized features |
> |$g_x$ | The real gradients of the quantized features |
> |$p$ | The weight vector of clustering centroid|
> |$f(\cdot)$ | The loss function |
> |$\Vert \cdot \Vert_2$ | The $\ell_2-$norm |
>
>
> **Q2:** Since additional fully connected layers are added to learn the channel information, how about the increased computation cost when the number of channels and devices is high?
> **A2:** Yes, controlling the computational cost is important to deploy SGQ on realistic edge devices. We have holistically inspected the average computational time proportion (%) of different SGQ components in Table 2 in the main submission. The cost of fully connected layers mentioned here is measured by the Channel-attention Grouping (CAG) block. The CAG block involves both forward and backward passes as its FC layers require parameter updating. The CAG block holds slight overhead in these two stages (**less than 2%**) and is independent to quantization bits, thus providing good extensibility to general CNN models. Overall, the computational cost of SGQ (especially the fully-connected layers of the CAG block) is controlled in an acceptable range that matches the on-device computational capacities.
>
> Meanwhile, during the design of the CAG block, we have considered the case of large channel numbers. Actually, a feature map of a great large channel number is beneficial to SGQ to obtain a better quality-size tradeoff, i.e., getting a higher compression ratio and model accuracy over existing methods. In our experiments, we covered extensive CNN architectures, where the channel number varies from **64 to even 512**. Such a degree of channel numbers corresponds to the setting of most modern CNNs. The evaluation results have shown that SGQ can consistently achieve great compression ratios while not dropping the model accuracy.
>
> In practice, we suggest using 4-bit quantization to make a balance between system overhead and model accuracy. In such a setting, our SGQ method could improve the image processing speed (images/sec) and achieve a good speedup **from 9.22× to 11.37×**, on average, over the baselines. Here, we present the inspection results in the following table.
>
>
> **Table: Average system overhead proportion (%) of computational time in different SGQ modules.**
> | # Bits                 |      SGQ | CAG's Forward Pass | CAG's Backward Pass | Gradient Calibration (GC) |    Total |
> |------------------------|---------:|-------------------:|--------------------:|--------------------------:|---------:|
> | 8 Bits                 |    31.93 |               1.09 |                1.88 |                      8.67 |    43.57 |
> | 6 Bits                 |    14.86 |               1.06 |                1.76 |                      4.51 |    22.19 |
> | **4 Bits (Suggested)** | **5.51** |           **1.05** |            **1.75** |                  **1.62** | **9.89** |
> | 2 Bits                 |     4.12 |               1.04 |                1.73 |                      1.24 |     8.14 |

---

### Official Review · Reviewer_jfAG · 2022-07-11

**Rating:** 6
**Confidence:** 2
**Soundness:** 2 fair
**Presentation:** 2 fair
**Contribution:** 2 fair

**Summary:**

CL system has the problem of communication deficiency caused by limited bandwidth.
Other solution (model slicing, matrix factorization) has may degrade the model accuracy if compressing features.
This paper propose new quantization scheme, called SGQ (Stripe-wise Group Quantization).
SGQ captures both channel and spatial-level similarity in pixels, and hierarchically encodes the features in these two levels to achieve a much higher compression ratio.

**Questions:**

Why didn't you experiment with large dataset (ex. ImageNet)?

**Limitations:**

CL system has the problem of communication deficiency caused by limited bandwidth.

**Strengths And Weaknesses:**

*strengths

The idea is straightforward, and the paper is easy to follow.
The observation of motivation is good, and the solution makes sense.

*weaknesses

In main paper, Cifar-10, Cifar-100  and mini-ImageNet are too small datasets.

---

> ### Author Response · Authors · 2022-08-01
> **Author Response to Reviewer jfAG (Part II)**
>
> We add holistic experiments to inspect SGQ's performance on ImageNet-1K.
> Here are four tables to report the detailed results on ImageNet-1K, in terms of model accuracy (%), traffic size (MB), inference speed (ms) and image processing speed (images/sec).
>
> **Table1: Comparison of Model Accuracy (%).**
> | Method                  | ResNet50 Acc. (%) | ResNeXt101 Acc. (%) | MobileNet-V3 Acc (%) | EfficientNet Acc. (%) |
> |-------------------------|------------------:|--------------------:|---------------------:|----------------------:|
> | FP32 (Acc. Upper Bound) |             75.56 |               77.17 |                72.13 |                 81.47 |
> | UP                      |             69.34 |               70.68 |                54.93 |                 61.86 |
> | PQ                      |             72.25 |               72.99 |                60.84 |                 67.59 |
> | CLIO                    |             11.89 |               12.46 |                10.18 |                 13.23 |
> | **SGQ**                 |         **74.67** |           **75.43** |            **70.39** |             **80.15** |
>
> **Table2: Comparison of Traffic Size (MB).**
> | Method  | ResNet50 Traffic Size (MB) | ResNeXt101 Traffic Size (MB) | MobileNet-V3 Traffic Size (MB) | EfficientNet Traffic Size (MB) |
> |---------|---------------------------:|-----------------------------:|-------------------------------:|-------------------------------:|
> | FP32    | 143.11                     | 169.63                       | 108.61                         | 85.22                          |
> | UP      | 36.60                      | 43.38                        | 27.78                          | 21.80                          |
> | PQ      | 26.35                      | 33.32                        | 18.37                          | 17.85                          |
> | CLIO    | 55.50                      | 66.24                        | 33.33                          | 37.03                          |
> | **SGQ** | **10.54**                 | **13.33**                   | **8.82**                      | **7.25**                      |
>
> **Table3: Comparison of Inference Speed (ms).**
> | Method  | ResNet50 Inference Speed (ms) | ResNeXt101 Inference Speed (ms) | MobileNet-V3 Inference Speed   (ms) | EfficientNet Inference Speed   (ms) |
> |---------|------------------------------:|--------------------------------:|------------------------------------:|------------------------------------:|
> | FP32    | 124.10                        | 147.10                          | 94.19                               | 73.90                               |
> | UP      | 40.89                         | 48.47                           | 31.04                               | 24.35                               |
> | PQ      | 32.89                         | 40.61                           | 23.69                               | 21.27                               |
> | CLIO    | 55.66                         | 66.33                           | 35.37                               | 36.25                               |
> | **SGQ** | **20.54**                    | **24.99**                      | **16.22**                          | **12.99**                          |
>
>
> **Table4: Comparison of Processing Throughput (images/sec).**
> | Method  | ResNet50 Processing Throughput (images/sec) | ResNeXt101 Processing Throughput (images/sec) | MobileNet-V3 Processing Throughput (images/sec) | EfficientNet Processing Throughput (images/sec) |
> |---------|--------------------------------------------:|----------------------------------------------:|------------------------------------------------:|------------------------------------------------:|
> | FP32    | 8                                           | 7                                             | 11                                              | 14                                              |
> | UP      | 24                                          | 21                                            | 32                                              | 41                                              |
> | PQ      | 30                                          | 25                                            | 42                                              | 47                                              |
> | CLIO    | 18                                          | 15                                            | 28                                              | 28                                              |
> | **SGQ** | **49**                                     | **40**                                       | **62**                                         | **77**                                         |
>
> The above four tables present a comprehensive inspection when adopting SGQ to the large-scale ImageNet. We can observe that SGQ significantly outperforms the existing methods in different metrics and achieves a good quality-traffic trade-off.

---

> > ### Comment · Reviewer_jfAG · 2022-08-09
> > **Thank you.**
> >
> > Thank you. I like the idea in this paper, and the application that this work potentially can bring to the community. The authors somehow could respond to my concerns so I vote for weak accept. I hope the author definitely adds the imagenet-1k experiment data to the appendix.

---

> > > ### Author Response · Authors · 2022-08-09
> > > **Thanks for Reviewer's Acknowledgment**
> > >
> > > Dear reviewer jfAG,
> > >
> > > Thank you very much for your kind comments and suggestions. We are sincerely encouraged by your acknowledgment of our work. In the final version, we will add detailed ImageNet-1K results to the appendix and try our best to improve the paper.
> > >
> > > Thank you for your time again!
> > >
> > > Best,
> > >
> > > Authors

---

> ### Author Response · Authors · 2022-08-02
> **Author Response to Reviewer jfAG (Part I)**
>
> We thank the reviewer for the constructive comments and hope to address your concerns as follows.
>
> **Q1:** About experiments with a large dataset (ex. ImageNet).
>
> **A1:** Considering the objective of establishing communication-efficient collaborative learning on edge devices, our evaluations follow the typical experimental setting adopted by previous works [1-5], including the configurations of comparison baselines, datasets, neural network models, deployment devices, and bandwidth environment. The dataset of CIFAR-10/100 and mini-ImageNet mentioned here are standard datasets used in the evaluation of previous works.
>
> Meanwhile, the edge devices usually hold much lower computational and storage capacity over the conventional GPU-based machines in the cloud. Such a resource-constrained environment also limits the choice of evaluation datasets. It is impractical if we directly deploy the original ImageNet-1K (1.2 million images with more than 140 GB storage demands) dataset on the commodity edge device (e.g., the typical NVIDIA Jetson Nano with 4GB RAM and 64 GB micro SD storage). Actually, the mini-ImageNet is a 100-class subset of ImageNet for one-shot learning and is more suitable for the evaluation based on the tiny edge devices. In real-world scenarios, edge devices often hold quite limited labeled data but we still want them to handle the edge intelligence applications [2-4]. The datasets used in our experiments can well match the realistic demands of collaborative learning on edge devices.
>
> Actually, SGQ can support the training on large-scale datasets if we are equipped with powerful machines. We verify this point by conducting the experiments on the ImageNet-1K datasets with ResNet50, ResNeXt101, MobileNetV2 and EfficientNet. As to the hardware configuration, We use a machine with NVIDIA 3090 GPU to replace the original edge device with the tiny Jetson Nano board, so as to provide sufficient computational and storage capacity for handling ImageNet. Other experimental configurations are the same as our main submission. Under the unified setting of using 8-bit compression, we observe that SGQ **consistently outperforms all the baselines in terms of model accuracy (%), traffic size (MB), inference speed (ms) and image processing throughput (images/sec)**. These results verify that our SGQ achieves a **good accuracy-traffic tradeoff** and well matches the learning requirements even using the large-scale ImageNet dataset. The detailed results are summarized in four tables in Author Response Part II.
>
> **References:**
> [1] BS-MCVR: Binary-sensing based Mobile-cloud Visual Recognition, In Proceedings of ACM International Conference on Multimedia (MM), 2021.
> [2] CLIO: Enabling automatic compilation of deep learning pipelines across IoT and Cloud, In Proceedings of Annual International Conference on Mobile Computing and Networking (MobiCom), 2020.
> [3] Octo: INT8 Training with Loss-aware Compensation and Backward Quantization for Tiny On-device Learning, In Proceedings of USENIX Annual Technical Conference (ATC), 2021.
> [4] Group Knowledge Transfer: Federated Learning of Large CNNs at the Edge, In Proceedings of Annual Conference on Neural Information Processing Systems (NeurIPS), 2020.
> [5] Layer-wised Model Aggregation for Personalized Federated Learning, n Proceedings of Annual Conference on Neural Information Processing Systems (NeurIPS), 2021.

---

> ### Author Response · Authors · 2022-08-09
> **Look Forward to Post-rebuttal**
>
> First and foremost, thank you very much for spending time providing constructive and valuable comments on our paper. We have carefully replied to your questions and presented the supplementary experiments on the **large-scale ImageNet-1K** datasets.
>
> These results verify that the proposed SGQ method can significantly outperform the existing methods in terms of **model accuracy (%), traffic size (MB), inference speed (ms) and image processing throughput (images/sec)**. We also provided more detailed **alation studies** to inspect how each part of SGQ impacts the model accuracy, and compare the **system overhead** proportion (%) of computational time on each part. The evaluation shows that SGQ consistently achieves a **great tradeoff between accuracy, latency and throughput** under different model scales.
>
> Since the discussion period is quickly drawing to a close, we would like to know if there are any additional clarifications or experiments that we can offer. We are looking forward to your feedback and sincerely invite you to update the score if concerns are addressed.
>
> Thanks a lot for your time again!

---

### Author Response · Authors · 2022-08-07
**Author General Response**

We sincerely appreciate all reviewers’ efforts in reviewing our paper and for the constructive feedbacks. Here, apart from the response to each reviewer, we would like to thank reviewers for the acknowledgment of our work and highlight new results added during the rebuttal.

We are encouraged that the reviewers appreciate and recognize our contributions:

* A novel work to establish a communication-efficient collaborative learning paradigm on commodity edge devices, striking a good balance between model accuracy and traffic saving [ofcH, B9sp, HVqg ].
* A practical solution with complete theoretical analysis and an efficient convergence rate for real-world applications [ofcH, B9sp].
* Support realistic collaborative learning procedures under extensive CNN models and compression bits via a novel stripe-wise quantization method [ofcH, B9sp, HVqg].
* Detailed experiments to verify the superiority of higher model accuracy and compression ratios over existing methods [ofcH, B9sp, HVqg].
* Well-written and easy to understand, with good motivation and clear methodology [jfAG, ofcH, B9sp, HVqg ].


In this rebuttal, we have added more supporting results following the reviewers’ suggestions.

* We thoroughly inspect the proposed SGQ under the **large-scale ImageNet-1K** dataset, including model accuracy (%), traffic size (MB), inference speed (ms) and image processing throughput (images/sec) [jfAG].
* We present the holistic **tradeoff** between accuracy, latency and throughput under different model scales, verifying that **SGQ significantly outperforms existing methods in these metrics** [B9sp, HVqg].
* We conduct more detailed **alation studies** on how each part of SGQ impacts the model accuracy, under different typical CNN models [B9sp, HVqg].
* We inspect the average **system overhead** proportion (%) of computational time of different SGQ modules, showing the effectiveness of our strip-wise quantization, Channel-attention Grouping (CAG) block and gradient calibration [ofcH].

Thank you for your time again!

---

### Author Response · Authors · 2022-08-07
**Author General Response**

Dear AC and reviewers:

Thanks again for your constructive comments, which have helped us a lot to improve the paper's quality and clarity.

Since the discussion period is quickly drawing to a close, please do not hesitate to let us know if there are any additional clarifications or experiments that we can offer. We would love to convince you of the merits of the paper and appreciate your suggestions.

Thanks for your time!

---

### Meta-Review · Area_Chair_bhDH · 2022-08-28

**Recommendation:** Accept
**Confidence:** Less certain

**Metareview:**

This paper proposes a novel communication-efficient learning method that significantly reduces feature size and communication traffic. The rebuttal solved the reviewers concerns about the dataset size and accuracy / latency trade off.

**Award:**

No

---

### Decision · Program_Chairs · 2022-09-14

Accept